# N-acetylglucosamine Signaling: Transcriptional Dynamics of a Novel Sugar Sensing Cascade in a Model Pathogenic Yeast, *Candida albicans*

**DOI:** 10.3390/jof7010065

**Published:** 2021-01-19

**Authors:** Kongara Hanumantha Rao, Soumita Paul, Swagata Ghosh

**Affiliations:** 1National Institute of Plant Genome Research, Jawaharlal Nehru University Campus, New Delhi 110067, India; 2Central Instrumentation Facility, Division of Research and Development, Lovely Professional University, Phagwara, Punjab 144411, India; 3Department of Molecular Biology and Biotechnology, University of Kalyani, Kalyani, West Bengal 741235, India; soumitapaul@klyuniv.ac.in (S.P.); ghosh_4@yahoo.co.in (S.G.)

**Keywords:** *Candida albicans*, transcription factor, GlcNAc, Ngs1, Rep1, Ron1, Hxk1, GlcNAc catabolism, signaling, histone acetylation, *Escherichia coli* (*E. coli*)

## Abstract

The amino sugar, N-acetylglucosamine (GlcNAc), has emerged as an attractive messenger of signaling in the pathogenic yeast *Candida albicans*, given its multifaceted role in cellular processes, including GlcNAc scavenging, import and metabolism, morphogenesis (yeast to hyphae and white to opaque switch), virulence, GlcNAc induced cell death (GICD), etc. During signaling, the exogenous GlcNAc appears to adopt a simple mechanism of gene regulation by directly activating Ngs1, a novel GlcNAc sensor and transducer, at the chromatin level, to activate transcriptional response through the promoter acetylation. Ngs1 acts as a master regulator in GlcNAc signaling by regulating GlcNAc catabolic gene expression and filamentation. Ndt80-family transcriptional factor Rep1 appears to be involved in the recruitment of Ngs1 to GlcNAc catabolic gene promoters. For promoting filamentation, GlcNAc adopts a little modified strategy by utilizing a recently evolved transcriptional loop. Here, Biofilm regulator Brg1 takes up the key role, getting up-regulated by Ngs1, and simultaneously induces Hyphal Specific Genes (HSGs) expression by down-regulating *NRG1* expression. GlcNAc kinase Hxk1 appears to play a prominent role in signaling. Recent developments in GlcNAc signaling have made *C. albicans* a model system to understand its role in other eukaryotes as well. The knowledge thus gained would assist in designing therapeutic interventions for the control of candidiasis and other fungal diseases.

## 1. Introduction

*Candida albicans* lives as a commensal on healthy individuals and can cause life-threatening systemic infections, contributing substantially to morbidity and mortality in immunocompromised patients [1,2,3]. The ability to utilize alternative carbon sources and yeast-hyphal morphogenetic flexibility are believed to be very critical for survival, host niche establishment, and virulence of the pathogen [4,5,6,7].

Accumulated bodies of evidence suggest that the amino sugar, N-acetylglucosamine (GlcNAc), not only acts as a very good alternative carbon and nitrogen source [4,8], but also as a potent inducer of signaling to stimulate expression of its own catabolic genes [9,10,11] and morphogenetic transitions, including yeast, to hyphal transition [12,13,14,15], and white-opaque switching, an epigenetic transition that controls mating process [16,17], and GlcNAc induced cell death (GICD) [18], a phenomenon that is characterized by the accumulation of reactive oxygen species (ROS), followed by rapid cell death via both apoptotic and necrotic mechanisms (GICD), and probably occurs due to GlcNAc triggering of important cellular processes. Therefore, it is important to understand the GlcNAc regulated cellular signaling for designing new anti-fungal therapeutic reagents to control candidiasis.

Studies on GlcNAc signaling in *C. albicans* have provided new insight into the potential role of GlcNAc in evoking dynamic cellular responses, and that made *C. albicans* a model system to study GlcNAc signaling in eukaryotes instead of *Saccharomyces cerevisiae* and *Schizosaccharomyces pombe*, which lack the genes required for GlcNAc utilization [19]. The mechanistic details of GlcNAc signaling are still under investigation in *C. albicans*.

Keeping in mind the multifarious role of the sugar, there is a pressing need to understand the GlcNAc triggered molecular mechanism of signaling, along with various transcription factors and proteins involved in this process. In the present review, we have discussed several research reports on GlcNAc signaling followed by the role of various transcription factors that evoke transcriptional responses, leading to operation of various cellular processes, such as GlcNAc metabolism and morphogenesis. Thereafter, we compared the mode of transcriptional regulation between *C. albicans* (eukaryote) and *Escherichia coli* (prokaryote), representing differences in GlcNAc signaling in terms of fundamental logic of gene expression. In the current review, we also narrated recent updates of accumulated knowledge to better explain the signaling network cascade.

## 2. A Brief History of GlcNAc Signaling Developments

N-acetylglucosamine (GlcNAc) is an often encountered carbon source in nature, being a component of peptidoglycan of the bacterial cell wall, fungal chitin, insect exoskeleton, and mammalian extra cellular matrix [20,21]. Apart from structural functions, its role in stimulating cellular signaling in pathogenic yeast *C. albicans* makes it an attractive sugar signaling molecule. The potential signaling role of GlcNAc to induce its own catabolic enzyme, N-acetylglucosamine kinase (Hxk1), was initially reported in 1974 [9]. Simultaneously, in another report, GlcNAc was shown to induce yeast-hyphal morphogenetic transition [12]. Then, onwards, GlcNAc induction system in *Candida albicans* has been used as a simple eukaryotic model system to study sugar (GlcNAc) induced signaling mechanisms of catabolic gene expression, and yeast, to hyphal morphogenetic transition [8,22,23,24].

Despite the fact that immobilized-GlcNAc (Agarose-GlcNAc) has the ability to trigger germ tube formation at the cell surface, observations from these studies were not appreciated due to lack of genetic approaches [10,24]. Results from the same group have simultaneously suggested that the induction of GlcNAc catabolism is uncoupled from the GlcNAc induced germ tube transition [10,25]. As technology improved, the advent of advanced tools facilitated molecular cloning and analysis of GlcNAc catabolic genes [26]. During further characterization of GlcNAc catabolic genes, it was shown that the genes responsible for GlcNAc catabolism, *NAG1*, *DAC1*, and *HXK1* exist as a cluster in the genome of *C. albicans*, and are regulated by a bidirectional promoter, wherein GlcNAc inducible factors bind at specific binding elements to induce gene expression [27]. Later, the GlcNAc catabolic pathway mutant analysis revealed the significance of GlcNAc catabolism for providing virulence in a mouse model of candidiasis and the role of GlcNAc kinase in hyper filamentous morphology [4]. Afterwards, a differential proteomics study for glucose vs. GlcNAc grown cells helped in the discovery of a novel GlcNAc specific transporter (Ngt1) that became the first eukaryotic GlcNAc transporter to be identified [28]. The *ngt1* mutant analysis further revealed that the entry of GlcNAc into cells induces GlcNAc responsive filamentation. Thereafter, it turned out to be a new tool for studying GlcNAc signaling mechanisms in *Candida* as well as other eukaryotes [29,30].

The internalized GlcNAc has the ability to directly induce transcriptional responses necessary for GlcNAc catabolism and hyphal initiation, while GlcNAc metabolism is not required for GlcNAc induced signaling [15,31]. Analysis of signaling mechanism in *hxk1* mutant background revealed that sensing of free non-phosphorylated GlcNAc is an advantageous character for organisms to differentiate external, non-phosphorylated GlcNAc from de-novo synthesized GlcNAc-6-phosphate, a precursor of Uridine diphosphate-GlcNAc (UDP-GlcNAc) that is constantly required for cell anabolic processes (Figure 1). This distinct feature increases signaling sensitivity so that the organism is able to detect minute levels of GlcNAc against internally synthesized GlcNAc-6-phosphate [15,32]. However, this mechanism does not appear to involve O-GlcNAcylation of proteins, a well conserved mechanism of signaling reported in several eukaryotes, including mammals and plants [33]. Analysis of *hxk1* mutant showed some interesting results, such as the tendency of the mutants to form filaments and induce the GlcNAc metabolic gene (*NGT1*, *NAG1*, *DAC1*, *GAL10*, etc.) expression under non-inducing conditions, such as glucose or galactose instead of GlcNAc. This is likely due to the accumulation of GlcNAc inside the cell, which could not be metabolized, and could induce signaling (15). Another possibility could be that Hxk1 acts as a repressor of yeast hyphal transition and GlcNAc metabolic gene expression (34). Further investigations are required to understand the molecular mechanism of signaling in detail.

In the subsequent studies, to uncover the pathway activated by GlcNAc, the genetic screen conducted using *C. albicans* deletion mutant library, helped in the finding of GlcNAc sensor Ngs1, which transduces signals just by directly binding to the free GlcNAc to induce transcriptional responses in the nucleus [34]. This has become a breakthrough finding in unraveling the molecular mechanism of GlcNAc signaling to control GlcNAc catabolic gene expression, as well as GlcNAc induced filamentation [34]. For stimulating hyphal morphogenesis/Hyphal Specific Genes (HSGs), Ngs1 adopts a slightly modified strategy that involves Biofilm regulator *BRG1* and hyphal repressor transcription factor *NRG1* [14,35], the mechanistic details of which were discussed in subsequent sections.

GlcNAc induces the expression of its own catabolic genes, viz; GlcNAc kinase (*HXK1*), GlcNAc-6-phosphate deacetylase (*DAC1*) and Glucosamine-6-phosphate deaminase (*NAG1*) and undergoes sequential action of degradation by Hxk1, Dac1, and Nag1, and finally enters into the glycolytic pathway [27]. In the first step, GlcNAc kinase converts GlcNAc to GlcNAc-6-phosphate (GlcNAc-6-P). In the second step, GlcNAc-6-P is converted to glucosamine-6-phosphate (GlN-6-P) with the removal of acetyl group by Dac1, and in the final step, GlN-6-P is converted to fructose-6-phosphate (fructose-6-P) with the removal of the amino group by Dac1. Eventually, fructose-6-P enters into the glycolytic pathway for providing energy and carbon source.

## 3. Role of Transcription Factors

GlcNAc signaling in *Candida albicans* involves a GlcNAc transporter Ngt1, a GlcNAc sensor, and transducer Ngs1, and an Ndt80 transcription factor Rep1 to induce the GlcNAc induced catabolic transcriptional responses [34]. The GlcNAc sensor, Ngs1, acts as a master regulator of GlcNAc signaling in *Candida albicans*, which senses GlcNAc and further triggers the promoter acetylation of GlcNAc catabolic genes to induce gene expression. Rep1 is an important Ndt80-family transcriptional factor that contains DNA Binding Domain (DBD) [34], which probably has specific binding sites on promoters of GlcNAc catabolic genes to recruit Ngs1 to cognate gene promoters (Figure 2). Thus, GlcNAc stimulates various cellular processes involving catabolism, transport, sugar scavenging, hyphal transition, virulence, and several uncharacterized transcript regulations through Ngs1 modulated transcriptional responses (Figure 2).

Both Ngs1 and Rep1 are constitutively expressed and found to be associated with GlcNAc catabolic gene promoters in GlcNAc independent manner, but their activation is GlcNAc dependent [34]. It has been reported that Ngs1 has two domains, the N-terminal highly conserved 3-Glycoside Hydrolase (GH_3_) domain, which is related to the bacterial β-N-acetylglucosaminidase domain [36], which manifests distinctive GlcNAc specificity [34]; and the other being the C-terminal GCN5 related N-acetyl transferase (GNAT) domain, which possesses an inherent histone acetylase property [37,38]. These two functionally different domains [34] take part in GlcNAc induced promoter activation, acetylation of chromatin, and further transcription of different GlcNAc metabolic genes.

Ngs1 also triggers GlcNAc induced filamentation by inducing *BRG1* expression. Unlike reports of previous studies [39], the cAMP/protein kinase A (PKA) pathway does not have a role in induction of GlcNAc stimulated filamentation of log phase cells under conditions of without fresh medium inoculation at 37 °C [40]. Su et al. have demonstrated a mechanism for the stimulation of yeast to hyphal transition under these conditions, without fresh medium inoculation in log phase cells at 37 °C [14]. For GlcNAc triggered hyphal formation, N-acetylglucosamine sensor, Ngs1, adopts a little modified strategy rather than directly targeting Hyphal Specific Genes (HSGs) promoters, although it employs the same promoter acetylation phenomenon as it does at other GlcNAc catabolic gene promoters (Figure 3, left panel). Most interestingly, Ngs1 mediates Hyphal Specific Genes (HSGs) expression by directly inducing the transcription of *BRG1* through promoter acetylation (Figure 3, right panel). Brg1 is a GATA (GATA DNA sequence) family transcription factor (Gat2) that was originally discovered as a Biofilm Regulator 1 (*BRG1*) [41]. It is known for filamentation induction and acts as a transcriptional regulator that works through Tor1 signaling pathway [14,42] in response to various environmental factors, such as serum [35,43,44].

Brg1 seems to play critical role in Ngs1 mediated hyphal transition by not only triggering the down regulation of *NRG1*, a global repressor of filamentation [45], but also modulating the chromatin remodeling of Hyphal Specific Genes (HSGs) promoters. The recent study conducted by Su et al., 2018 [14], reveals the molecular mechanism of Brg1 mediated regulation of Hyphal Specific Gene (HSG) expression. Upon entry of GlcNAc, Brg1 is upregulated by the action of the GNAT domain of Ngs1. The histone acetyl transferase activity of the GNAT domain loosens the nucleosome interaction at the *BRG1* promoter that results in the up-regulation of *BRG1*. Brg1 simultaneously down-regulates the expression levels of *NRG1* in log phase cells under the condition of being without fresh-medium inoculation (Figure 4A) [14]. Brg1 also plays a pivotal role for maintenance of filamentation in fresh-medium inoculation conditions in which activated Brg1 down regulates *NRG1* and also recruits Hda1, a histone deacetylase to the promoters of Hyphal Specific Genes (HSGs) to achieve nucleosome re-positioning, thereby, masking the DNA binding site of Nrg1 [14,42,46] (Figure 4B). GlcNAc induced hyphal induction and maintenance thus occurs through Brg1 mediated down-regulation of Nrg1 expression, which is initiated by GlcNAc sensing by Ngs1. As shown in Figure 4A under GlcNAc hyphal inducing conditions, Brg1 down regulates *NRG1* expression; similarly, over-expression of *NRG1* under regulatable *MET3* promoter appears to repress *BRG1* expression in a GlcNAc dependent manner [14]. Thus, Brg1 controls filamentation through a feedback loop, while Nrg1 represses *BRG1* transcription. Brg1 down-regulates *NRG1* expression levels by interfering with its mRNA stability [47]. This entire mode of action has been schematically represented in Figure 4A.

Thus, the recent discovery of new regulatory loops of transcription assisted in understanding and unraveling of signaling cascades that might be operating under in vivo conditions. The studies from Haoping Liu’s [14] group have provided new insights into the understanding of filamentation signaling pathway triggered by various environmental cues, including GlcNAc, serum, neutral pH, and temperature that are encountered in host niches. They have demonstrated an underlying mechanism for the stimulation of yeast to hyphal transition in log phase cells at 37 °C without fresh-medium inoculation, where Ngs1 appears to be involved in the down regulation of *NRG1* expression instead of cAMP-PKA pathway. Ngs1 employs Brg1 for *NRG1* down regulation. GlcNAc activates Ngs1 to acetylate the *BRG1* promoter, just like catabolic gene promoters, to induce the expression [14].

Recently, several mutational investigation has manifested a cascade of transcription factors (TFs), which are involved in GlcNAc metabolism, and which play a crucial character in morphological switching as well as its virulence [48,49,50]. These studies show that the *NDT80* family is a group of transcription factors that help to form resistance against different stresses and also plays a pivotal role in development of hyphae and virulence in *Candida albicans* [51,52]. The study of this transcription factor family has become very important as it is highly conserved in fungi [53]. Some of the recent research developments discussed in this review indicate that these Ndt80 family members are fundamental activators of GlcNAc catabolic genes [49]. *NDT80*, *RON1*, and *REP1* are three paralog of Ndt80 in *Candida albicans*. Ndt80-like transcription factor, Rep1, is shown to be an essential for growth on GlcNAc and induction of GlcNAc catabolic genes [34]. It recruits GlcNAc sensor Ngs1 to the promoters of GlcNAc catabolic genes (Figure 2). It is also needed for growth on galactose apart from GlcNAc [34]. Initially, it was recognized as a negative regulator of *MDR1* expression, the drug efflux pump [51]. In *C. albicans*, in addition to Rep1, there is another Ndt80 domain transcriptional factor Ron1, which is also appeared to be needed for growth on GlcNAc [50]. Ngs1 and Rep1 appear to control the overlapping set of genes needed for GlcNAc utilization, and separate gene sets specific to each factor for utilization of respective sugars. Further work needs to be carried out to understand the detailed molecular aspects of the regulation in response to various sugars, including GlcNAc.

Apart from Ngs1-Brg1 mediated signaling, GlcNAc also triggers hyphal transition through the pH sensing pathway, where Rim101 acts as transcriptional regulator that is known to induce hyphal formation in response to alkaline pH, and also provides adaptation to high ambient pH [54]. Here, GlcNAc metabolism appears to play a role, where the amino group is cleaved by deaminase (Nag1) from glucosamine-6-phosphate, during GlcNAc, catabolism leads to the accumulation of ammonia in the cytosol. Simultaneously, it extrudes out (and results in) the alkalization of the external environment that triggers the Rim101 dependent pathway [55,56]. Thus, GlcNAc appears to induce the yeast hyphal transition by two signaling pathways—one that gets directly activated by GlcNAc binding to the GlcNAc sensor, Ngs1, at the promoter region [34], and another that results from catabolism of GlcNAc, which increases the extracellular pH and activates Rim101 for hyphal gene induction [55].

GlcNAc is also known to induce another morphological transition, referred to as white-opaque switching (17). White-opaque switching is an epigenetic phenomenon that occurs without a change in DNA sequence and is stochastic in nature. Opaque transition is an important prerequisite step for mating [16,57] wherein elongated or bean-shaped cells form mating projections. GlcNAc produced by bacteria in the gastrointestinal tract acts as a natural source to induce white to opaque switching in *C. albicans* [58]. Ras1-cAMP/PKA pathway is implicated in the GlcNAc induced opaque switching. Here, phosphorylation of Wor1, a master regulator [59] that modulates opaque specific transcriptional response, is critical for switching, and it works downstream of the Ras1-cAMP/PKA pathway [17,48]. As we discussed in the previous sections, interestingly, in contrast to white-opaque switching, yeast to hyphal transition triggered by GlcNAc appeared to be independent of the cAMP/PKA pathway in a condition specific manner (14,43,58). Thus, these two morphogenetic transitions, yeast to hyphal transition, and white to opaque switching, appear to be triggered by GlcNAc through different pathways.

## 4. Role of N-acetylglucosamine Kinase, Hxk1 in GlcNAc Signaling

N-acetylglucosamine Kinase (Hxk1) is the first enzyme involved in the GlcNAc catabolism that phosphorylates GlcNAc and is involved in GlcNAc priming, which diverts GlcNAc to anabolic reactions, as shown in Figure 1. Initial studies have indicated that *C. albicans* mutants, disrupted for the GlcNAc catabolic pathway, showed attenuated virulence in a mouse model of candidiasis [4], and surprisingly, *HXK1* disruption mutant exhibit hyper-filamentous morphology under filamentation inducing conditions [4]. Then, the studies conducted to understand the role of the GlcNAc catabolic pathway on GlcNAc induced signaling revealed that GlcNAc metabolism is not required for its signaling [15,31]. Although, GlcNAc catabolism does not have a role in signaling, the catabolic enzyme, GlcNAc kinase (Hxk1) appears to repress the expression of GlcNAc metabolic genes including *NGT1*, *NAG1*, *DAC1*, *GIG2*, etc., galactose catabolic genes *GAL10*, *GAL7*, *GAL1*, and other genes, such as *PGK1*, *ADH1*, etc., in the presence of unrelated sugars, such as glucose [31], hinting towards the role of Hxk1 as a transcriptional regulator. However, there are some interesting data that argue that the altered/increased GlcNAc metabolic gene expression and filamentation abnormalities noticed in the *hxk1* mutant are due to an increased amount of intracellular GlcNAc [15], as it cannot be catabolized—but not due to the putative transcriptional role of Hxk1. Further investigations need to be carried out to understand the detailed insights into the underlying mechanism.

Interesting findings on N-acetylglucosamine kinase Hxk1, from *Candida albicans*, show that the gene also takes part in regulating white to opaque switching and cell wall synthesis in addition to GlcNAc metabolic gene expression and filamentation [5,31,60]. Furthermore, dynamic changes in the subcellular localization of Hxk1 additionally reflects its multifunctional plasticity. In *Candida*, the putative role of Hxk1 in mediating signaling of GlcNAc metabolic gene expression is entirely unknown. The galactose sensory signaling that is mediated by galactokinase Gal3 (lacking galactokinase activity) has been studied in yeast *Saccharomyces cerevisiae*. It has been shown that upon sensing galactose, Gal3 binds directly to Gal80 repressor and controls Gal4 function to activate galactose catabolic gene expression [61]. However, in *C. albicans*, GlcNAc kinase Hxk1 seems to adopt repression mechanism for GlcNAc catabolic gene expression when cells are present in a non-preferred carbon source, and upon sensing GlcNAc, Hxk1 probably relieves GlcNAc catabolic gene promoters, which is yet to be investigated.

## 5. Differences in the Fundamental Logic of GlcNAc Catabolic Gene Expression between *C. albicans* and *E. coli*

The mode of GlcNAc catabolic gene induction in *E. coli* (prokaryotes) [62] is simple and different from *C. albicans* (eukaryote) (Figure 5). In *E. coli*, GlcNAc is converted to GlcNAc-6-phosphate by a Phosphotransferase System (PTS) that simultaneously imports and phosphorylates GlcNAc [63]. NagC is a repressor that binds at the promoter of GlcNAc catabolic genes. Upon binding to GlcNAc-6-phosphate, NagC is released from catabolic gene promoters so that catabolic genes are turned on (Figure 5B) [64,65]. Contrastingly, in *C. albicans*, as discussed above, non-phosphorylated exogenous free GlcNAc that is imported to nucleus bind to the GlcNAc sensor and transducer, Ngs1, to activate catabolic gene expression through the acetylation of promoters (Figure 5A) [34]. These fundamental differences in gene regulatory mechanisms for the stimulation of GlcNAc catabolic gene expression in both *E. coli* and *C. albicans* probably reflect fundamental differences in logic of gene expression that basically arise due to diversity in chromatin status between two evolutionarily distant organisms [66]. Eukaryotic chromatin exists in a higher order condensed state; so the acetylation of histone proteins result in loosening of chromatin, thereby making the promoter DNA accessible to RNA polymerase II, or other transcription factors, to initiate transcription. On the contrary, in prokaryotes, the chromatin is organized in a simple DNA packaging state where promoters exist in non-restrictive ground state. Simple relieving of repressors result in the promoter DNA accessibility to RNA Polymerase II for gene induction. Thus, prokaryotes and eukaryotes display a difference in the fundamental logic of gene expression [66].

As depicted in Figure 5A, DNA binding transcription factor Rep1 keeps GlcNAc sensor Ngs1 in a recruited state to GlcNAc catabolic gene promoters in a GlcNAc independent manner. Upon GlcNAc sensing by Ngs1, the H3K9 and H3K14 region of histone protein at GlcNAc catabolic gene promoters get acetylated by Ngs1. This chemical alteration neutralizes the positive charge of histone tails that result in loosening of histone-DNA interaction, which further allow different transcription factors to access the promoter chromatin to transcription (Figure 5A) [34]. In contrast, in *E. coli*, chromatin exists in a simple non-condensed and non-restrictive state; so, once repressor NagC is removed from the promoter DNA, the promoter is accessible for basal transcriptional machinery to initiate transcription (Figure 5B).

## 6. What Makes *Candida albicans* an Important Model System to Study GlcNAc Signaling?

The impacts of GlcNAc signaling are prominent and can be measured in the form of transcriptional responses and visualized/manifested through clear effects on growth and morphological transition [11,17,27,28,34] (yeast to hyphae and/or white to opaque switching).Being a simple eukaryotic system, it shares close lineage with general model yeasts *Saccharomyces cerevisiae* and *Schizosaccharomyces pombe* [67]. The completely sequenced genome [68], currently available sophisticated tools for gene manipulation and characterization, have made this system more amicable for study [69,70,71].GlcNAc regulates multifaceted cellular functions that make it very interesting to understand the signaling mechanisms [21,48].

## 7. Concluding Remarks and Future Perspectives

Accumulating evidences revealed that, in *C. albicans*, the amino sugar GlcNAc acts as an excellent signaling molecule that modulates dynamic cellular processes, including its own sugar metabolism, morphogenetic transition (yeast to hyphae and white to opaque switch), GICD, virulence, and inter-species communication, etc. [20,48]. Thus, it is believed that GlcNAc signaling and its utilization (metabolism) provides an adaptive advantage for *C. albicans* to respond rapidly and appropriately to the host niche environmental conditions for maximal use of nutrients and competing over other microbes [18]. Therefore, it is very critical to unravel the molecular mechanisms of GlcNAc signaling in this fungal pathogen. The imported non-phosphorylated GlcNAc has the ability to directly induce the signaling and its catabolism is not required for its signaling [15,31]. GlcNAc binds to a novel GlcNAc sensor, Ngs1, at catabolic gene promoters, and promotes gene expression by activating Ngs1 for acetylation of cognate promoters [34]. Most surprisingly, the cAMP-PKA pathway is not involved in GlcNAc induced filamentation for log phase cells under the conditions of being without fresh-medium inoculation at 37 °C. In GlcNAc induced filamentation, also, GlcNAc sensor Ngs1 is involved through a little modified strategy by regulating *BRG1* expression and inhibiting the inhibitory effect of Nrg1 for promoting filamentation (14). Interestingly, it appears that GlcNAc triggers white/opaque switching and yeast/hyphal transition contrastingly through two different pathways; while white/opaque switching occurs through the Ras1-cAMP/PKA dependent pathway, yeast/hyphal transition occurs independently [14,17,40,48,55].

Although it appears that Ngs1-Rep1 complex is directly involved in the induction of GlcNAc catabolic gene expression by acetylation of cognate gene promoters, possible involvement of other regulatory factors that might act as either repressors and/or activators in the induction mechanism (mentioned in open questions), cannot be ruled out. In support of this argument, the negative role of Hxk1 in GlcNAc induction mechanism can definitely gain considerable attention, for which the molecular mechanism is yet to be revealed [31]. Further, it is clearly understood from the recent unbiased studies that, the same sensor, Ngs1, is involved in the GlcNAc induced metabolic gene expression, as well as GlcNAc induced filamentation. As signaling of these two processes appears to occur independently in response to environmental conditions, it remains an important aspect of understanding how Ngs1 chooses its target promoters, according to specific stimulus (whether filamentation or catabolism or both together). Additionally, *NGS1* has been shown to be important for the growth on maltose.

As GlcNAc sensor Ngs1 and associated Ndt80 transcription factor Rep1 are constitutively bound to promoter regions, it is important to determine the genome wide occupancy of these factors. Either ChIP-seq or ChIP-on-chip can be used to find out genome-wide binding regions, as well as sequence of the binding elements [72]. This analysis coupled with differential RNA-seq analysis would reveal the Ngs1 and Rep1 binding on gene promoters and respective activated genes in response to specific carbon sources, including GlcNAc. As studies have shown that GlcNAc signaling is very important for virulence and survival of the pathogen, the accumulated knowledge would provide new insights into the development of therapeutic drug targets for control of candidiasis and other fungal diseases.

## 8. Open Questions

As it is known that GlcNAc induced filamentation is different from GlcNAc induced metabolic gene expression, how does GlcNAc differentiate these two cellular processes separately; what is the molecular mechanism, and are there any environmental factors influence these processes?Is there any role of Ngs1 in GlcNAc induced white to opaque switching?What is the mode of regulation for Hxk1 mediated GlcNAc signaling to stimulate metabolic gene expression?What are the genome-wide binding sites for Ngs1 and Rep1, and is there any role of these transcription factors for the growth on other carbon sources, such as, galactose, mannose, etc.?What is the DNA-binding transcription factor responsible for the recruitment of Ngs1 at the promoter region of *BRG1*?

## Figures and Tables

**Figure 1 jof-07-00065-f001:**
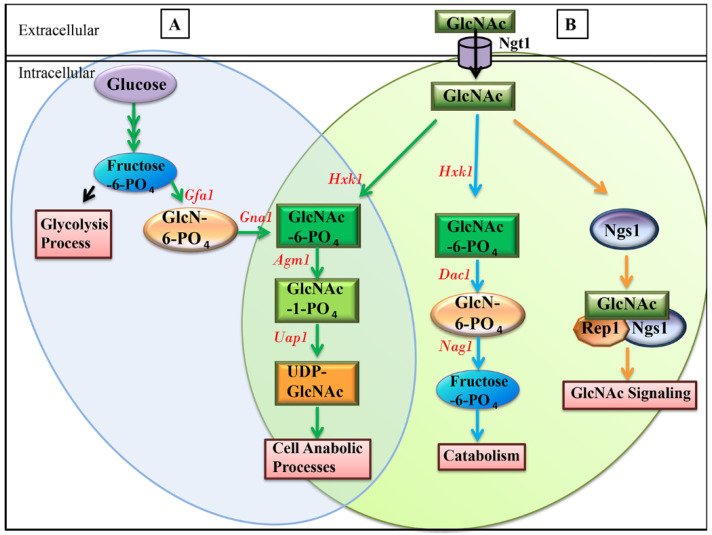
Direct import of external N-acetylglucosamine (GlcNAc) induces the signaling, but its metabolism is not required for GlcNAc signaling. Detection of external, non-phosphorylated GlcNAc provides sensitivity against internally synthesized GlcNAc-6-phosphate. (**A**) The internal de novo synthesized GlcNAc-6-P is utilized in the anabolic processes/functions, such as chitin synthesis, N-glycosylation, GPI-anchors, etc., and is not involved in the GlcNAc signaling pathway. De novo synthesis of GlcNAc-6-P involves sequential action of GlcN-6-P synthase (Gfa1) and GlcN-6-P acetyltransferase (Gna1) from fructose-6-P (Glycolysis). GlcNAc-6-P gets further converted to UDP-GlcNAc by the sequential action of phosphor acetylglucosamine mutase (Agm1) and UDP-GlcNAc pyrophosphorylase (Uap1) [32]. (**B**) The internalized GlcNAc through the GlcNAc specific transporter (Ngt1) can directly induce the signaling by binding to GlcNAc sensor (Ngs1). GlcNAc is also metabolized to GlcNAc-6-P by GlcNAc kinase (Hxk1), which is either catabolized using sequential action of Deacetylase (Dac1) and Deaminase (Nag1) to provide energy, or converted to UDP-GlcNAc involved in anabolic functions. The overlapping area between A and B represents the steps that are common to both the de novo pathway and imported GlcNAc pathway used for anabolic functions. Note: while steps involved in the anabolic pathways are represented with green arrows, the steps involved in the GlcNAc catabolic and GlcNAc signaling pathways are represented with blue and orange arrows, respectively.

**Figure 2 jof-07-00065-f002:**
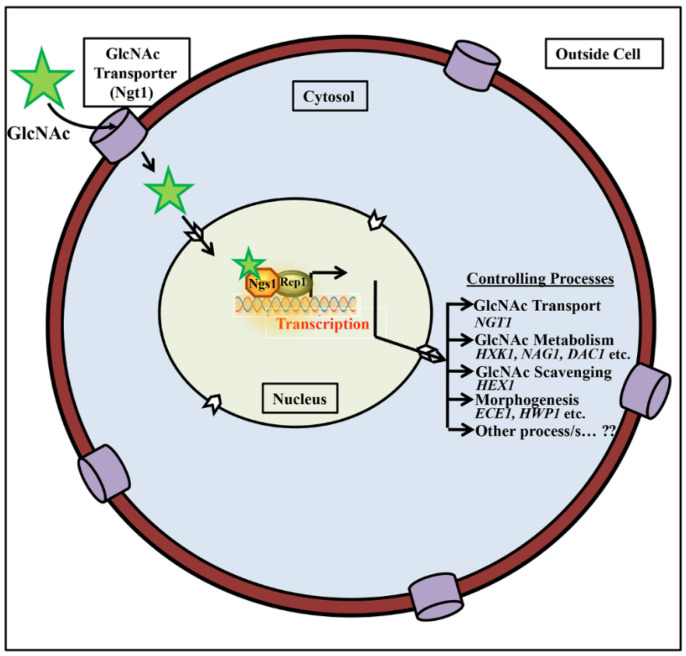
Working model for GlcNAc import and its signaling in *C. albicans*. Exogenous GlcNAc is imported into cytosol through GlcNAc specific transporter, Ngt1. In the nucleus GlcNAc binds to GlcNAc sensor, Ngs1 to activate transcription of genes involved in various processes (37) as shown in the figure. Transcription factors, such as Rep1, which bind to DNA, recruit GlcNAc sensor Ngs1 at cognate gene promoters and transduce GlcNAc signaling. Ngs1 acts as a master regulator of GlcNAc signaling by regulating various cellular processes, such as GlcNAc transport (*NGT1*), GlcNAc metabolism (*HXK1*, *NAG1*, *DAC1* etc.), GlcNAc scavenging (*HEX1*), morphogenesis (*HWP1*, *ECE1* etc.), and other uncharacterized functions.

**Figure 3 jof-07-00065-f003:**
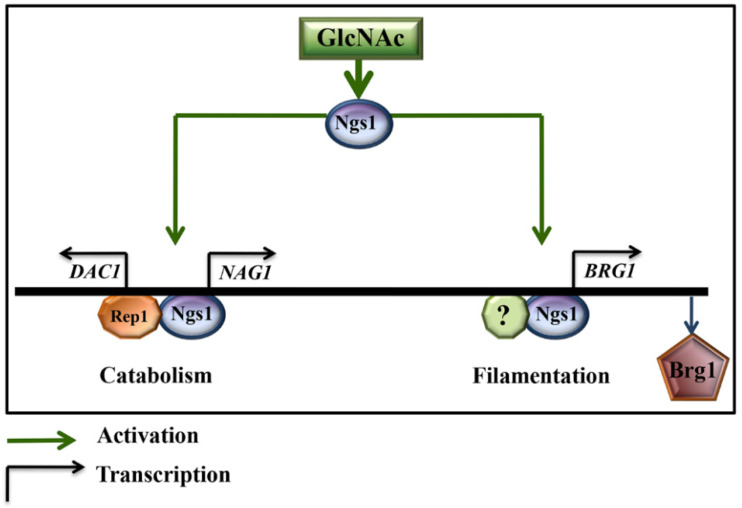
N-acetylglucosamine (GlcNAc) stimulates signaling via GlcNAc sensor and transducer Ngs1. Upon GlcNAc binding, Ngs1 induces expression of GlcNAc catabolic genes (e.g., *NAG1* and *DAC1*) [34] and filamentation regulating transcription factor, *BRG1* by acetylating at their promoters [14]. Transcription factor (TF) Rep1 recruits Ngs1 to the GlcNAc catabolic gene promoters (left panel), whereas TFs recruiting Ngs1 to *BRG1* promoter are yet to be found (right panel). The active relationship is represented with green lines and transcriptional promotion is represented with black arrows.

**Figure 4 jof-07-00065-f004:**
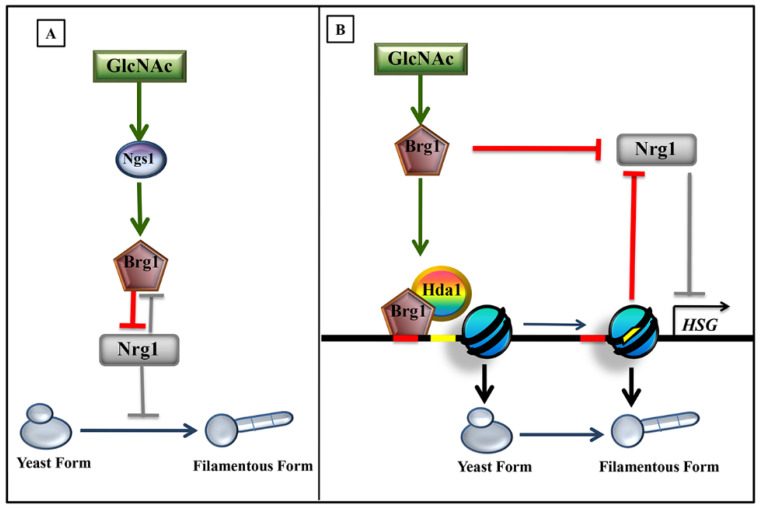
Regulation of N-acetylglucosamine (GlcNAc) induced filamentation occurs in two modes. (**A**) Upon GlcNAc binding/sensing, the activated GlcNAc sensor (Ngs1) upregulates/activates *BRG1* through its histone acetylase activity (37). The accumulated Brg1 down regulates *NRG1* repressor, and thereby induces Hyphal Specific Gene (HSG) expression to promote hyphal transition. This mechanism occurs in log phase cells at 37 °C under conditions without fresh-medium inoculation [14]. (**B**) Under inoculation conditions in presence of GlcNAc or serum, etc., the hyphal maintenance occurs through activation of *BRG1*. Brg1 down regulates *NRG1* expression and also causes nucleosome repositioning at Hyphal Specific Gene (HSG) promoters by interacting with histone deacetylase Hda1 to inhibit Nrg1 binding at regulatory sites [46]. The active relationship is represented with green and red lines, whereas inactive relationship is represented with grey lines.

**Figure 5 jof-07-00065-f005:**
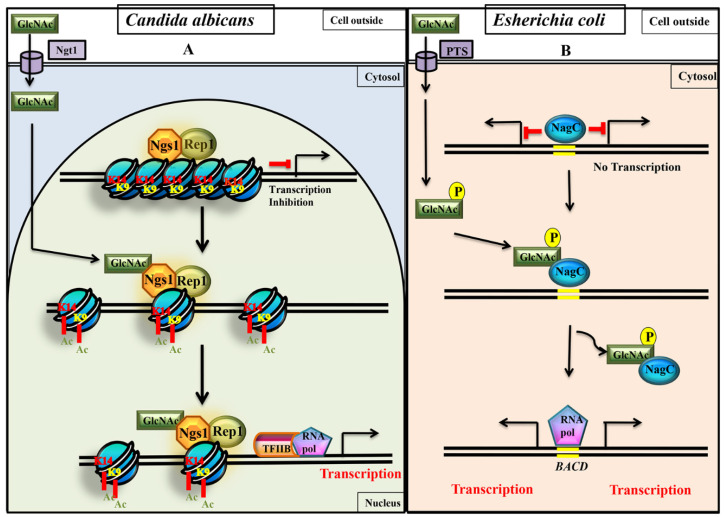
Comparison of GlcNAc signaling between eukaryotes (*Candida albicans*) and prokaryotes (*Escherichia coli*). During signaling process, in *C. albicans* the non-phosphorylated free GlcNAc directly binds to GlcNAc sensor (Ngs1) to promote histone H3 acetylation (H3K9 Ac and H3K14 Ac) of the nucleosomes at the promoters of GlcNAc catabolic genes [34], whereas in *E. coli*, GlcNAc-6-phospahte formed from Phosphotransferase System (PTS) binds to NagC repressor to release it from operator region of GlcNAc catabolic genes [63,65]. (**A**) Working model depicting induction of GlcNAc catabolic genes through Ngs1-Rep1 mediated acetylation. More work needs to be carried out to understand the role of other proteins involved in the determination of Ngs1 specificity for GlcNAc over the other carbon sources like maltose (53). The chromatin modifying activities of activators (Ngs1-Rep1 complex) lead to altered chromatin structure (loosening of nucleosomes) thereby transcription of cognate genes is promoted. (**B**) In prokaryotes, as the chromatin architecture is simple, just the removal of repressor NagC is sufficient to initiate transcription.

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
