# Peer review of "N-acetylglucosamine Signaling: Transcriptional Dynamics of a Novel Sugar Sensing Cascade in a Model Pathogenic Yeast, Candida albicans"

_jof, 2021, doi:10.3390/jof7010065_

Round 1
Reviewer 1 Report
With the revision and resubmission, the quality of the manuscript “N-acetylglucosamine Signaling: Transcriptional dynamics of a novel Sugar Sensing Cascade in a model pathogenic yeast, Candida albicans” has been improved.
However, I still do not see the need to put captions such as Figure 1 etc. within the figures, after all the caption is also under the figure. Moreover, there are still editorial errors, differences in the formatting of individual subsections and I strongly suggest reviewing the manuscript by a native speaker.
Author Response
Q1. With the revision and resubmission, the quality of the manuscript “N-acetylglucosamine Signaling: Transcriptional Dynamics of a Novel Sugar Sensing Cascade in a Model Pathogenic Yeast, Candida albicans” has been improved.
However, I still do not see the need to put captions such as Figure 1 etc. within the figures, after all the caption is also under the figure. Moreover, there are still editorial errors, differences in the formatting of individual subsections and I strongly suggest reviewing the manuscript by a native speaker.
Ans: We have removed captions like figure 1, figure 2 etc. from the figures. We have rectified editorial errors as far as possible throughout the text and changes are highlighted with yellow.
Reviewer 2 Report
The manuscript has been improved over the original version. However, as indicated below, there are still a lot of issues with the manuscript that need to be addressed and many sections of the manuscript are written unclearly. Thus, the manuscript needs major revisions in addition to the specific comments below.
- The authors should emphasize the role of Ngs1 in acetylating chromatin is suggested by the published data, but further work needs to be done. For example, the legend to Fig. 5 should indicate that it is a model, and that more experimental work needs to be done to verify this model. There are likely more proteins acting in this system to give specificity for GlcNAc since ngs1 mutants also fail to grow on other sugars.
- Line 22 and 142: Change Ndt80 to Ndt80-family
- Line 44-45. Not clear how death can provides ability to colonize nutrient poor niche
- Line 54. “with various players/transcription factors” is unclear and sounds like lab jargon.
- Line 65. Change “constitutional function”. This is not appropriate
- Line 167, 169, 193, 210, 221, 357, . What does this phrase mean: “in log phase cells under conditions of without inoculation at 37°C”? Without inoculation there would be no cells in the culture. Please explain.
- Lines 222 – 223. Rewrite unclear sentence: “instead of cAMP-PKA pathway does.”
- Lines 225-226.This sentence contradicts some previous statements in the review, and is also contradicted by the literature.
- a) Inoculation into fresh medium induces hyphal growth independently of GlcNAc.It is an inducer all by itself (see Enjalbert and Whiteway, 2005).
- b) The Ras-cAMP pathway is not needed for cells to induce transcriptional responses, since a cyr1Dmutant induces the GlcNAc catabolic genes very well (Gunasekera et al, 2010).
- c) The Ras-cAMP pathway is needed for cells to grow well, it is not needed for GlcNAc induce hyphal growth (Parrino et al, 2017).
- Line 232: change fungus to fungi
- Line 269: what is “GlcNAc priming.”?
- Line 270:Change “in mice model candidiasis” to “ in a mouse model of candidiasis”
- Line 318: Change “leads to neutralize” to “neutralizes”
- Line 334: delete “as”
- Section 6. I would also add a fourth point that GlcNAc can induce signaling independently of its metabolism. This permits it to be studied as a signaling molecule independent of cellular metabolism.
- Line 345: the commas are not needed
Author Response
The manuscript has been improved over the original version.However, as indicated below, there are still a lot of issues with the manuscript that need to be addressed and many sections of the manuscript are written unclearly.Thus, the manuscript needs major revisions in addition to the specific comments below.
1.The authors should emphasize the role of Ngs1 in acetylating chromatin is suggested by the published data, but further work needs to be done. For example, the legend to Fig. 5 should indicate that it is a model, and that more experimental work needs to be done to verify this model. There are likely more proteins acting in this system to give specificity for GlcNAc since ngs1 mutants also fail to grow on other sugars.
Ans: It is a valuable suggestion because the role of other proteins that determine the specificity of GlcNAc catabolic gene regulation is yet to be worked out as ngs1mutant also showed growth defects on other sugars like maltose at 37°C in addition to GlcNAc.In the revised version of the manuscript we have made modifications in the figure 5 legend as well as in the relevant text, and modifications are highlighted with yellow. Lines 334-337.
2.Line 22 and 142: Change Ndt80 to Ndt80-family
Ans: ‘Ndt80’ is changed to ‘Ndt80-family’, and the change is highlighted with yellow in lines22 and 144.
3.Line 44-45.Not clear how death can provides ability to colonize nutrient poor niche
Ans: GICD cannot provide the ability to Candida cells to colonize nutrient poor niche. GICDis characterized by accumulation of reactive oxygen species (ROS) followed by rapid cell death via both apoptotic and necrotic mechanisms (GICD) and is probably occurs due to GlcNAc triggered important cellular processes that provide ability to the pathogen tocolonize nutrient poor niche.Now in the revised version of the manuscript we have edited the sentence (Lines45 to 47) to make it clearly understandable.
4.Line 54. “with various players/transcription factors” is unclear and sounds like lab jargon.
Ans:Now we have edited the sentence. Lines 56-57.
25.Line 65. Change “constitutional function”. This is not appropriate
Ans:Now we have changed the word ‘constitutional’ with ‘structural’ in Line 67.
6.Line 167, 169, 193, 210, 221, 357, What does this phrase mean: “in log phase cells under conditions of without inoculation at 37°C”? Without inoculation there would be no cells in the culture. Please explain.
Ans:“in log phase cells under conditions of without inoculation at 37°C” means that the exponentially growing cells at 37°C are directly induced as such without re-suspending (inoculating) into a fresh medium. This method of induction is different from that routinely followed laboratory method wherein cells from saturated culture are re-suspended into a fresh inducing medium.
7.Lines 222 –223. Rewrite unclear sentence: “instead of cAMP-PKA pathway does.”
Ans:Now the sentence is rectified by removing “does” at the end, in line 225.
8.Lines 225-226.This sentence contradicts some previous statements in the review, and is also contradicted by the literature.a) Inoculation into fresh medium induces hyphal growth independently of GlcNAc. It is an inducer all by itself (see Enjalbert and Whiteway, 2005).
b) The Ras-cAMP pathway is not needed for cells to induce transcriptional responses, since a cyr1Dmutant induces the GlcNAc catabolic genes very well (Gunasekera et al, 2010).
c) The Ras-cAMP pathway is needed for cells to grow well, it is not needed for GlcNAc induce hyphal growth (Parrino et al, 2017).
Ans:To avoid the contradiction we have modified the sentence in the revised version of the manuscript. The sentence in lines 226-230 as, “But whereas under the conditions when cells from a saturated culture are inoculated into a fresh medium, GlcNAc triggered filamentation appears to be dependent on Ras1-cAMP-PKA pathway [42, 48, 51]. Still, there are some reports indicating the involvement of additional pathways that likely to regulate GlcNAc-induced filamentation [43,58].”
a)This paper explains about the inherent capacity of the growing culture cells in specific stage to induce hyphae irrespective of the inducers like serum or GlcNAc. They have shown that release from inhibition,caused by quorum-sensing molecule farnesol is responsible for induction of hyphae when stationary-phase cells are inoculated in to a fresh medium (Enjalbert and Whiteway, 2005). But to maintain sustained induction of filamentation inducers like GlcNAc, serum are required.
b)Overall, various studies conducted to understand the molecular mechanism of hyphal induction in stationary phase cells inoculated into fresh medium revealed that the considerable decrease in Nrg1 levels is critical to induce filamentatio which is achieved through Nrg1 degradation by the release from farnesol inhibition as well as NRG1down regulation by cAMP-PKA pathway activation at 37°C. in general, either of the pathways alone is not sufficient to reduce Nrg1 levels to induce hyphal specific genes required for hyphal induction and elongation (Lu et al., 2011; Lu et al., 2014-PNAS)
c)Whereas under the conditions without inoculation (in log phase),cAMP independent pathways operate and inducers like GlcNAc, serum and neutral pH can activate BRG1 which in turn down regulate NRG1expression that relieves filamentous specific gene expression (Su et al., 2018). Here, GlcNAc sensor Ngs1 activates BRG1 upon binding with GlcNAc.
d)Decrease in Nrg1 levels shown to be essential for filamentation.Recent studies have shown that cell stage plays an important role for inducing filamentation by responding through specific signaling pathways. For example Ngs1 is involved in log phase cells without inoculation while cAMP-PKA pathway and removal farnesol inhibition are involved in stationary phase cells inoculated into fresh medium for decreasing Nrg1 levels, and there by induce filamentation.
9.Line 232: change fungus to fungi
Ans:fungus is changed to fungi. Line 236.
10.Line 269: what is “GlcNAc priming.”?
Ans: Just like priming reaction that occurs in initial step of glycolysis for the conversion of glucose to glucose-6-phosphate with the utilizing ATP by hexokinase, GlcNAc is also converted to GlcNAc-6-phosphate by utilizing ATP energy, and this reaction is carried out by N–acetylglucosamine kinase. Priming helps in the driving the reaction towards forward direction.
11.Line 270:Change “in mice model candidiasis” to “ in a mouse model of candidiasis”
Ans: Now the phrase has been changed to “in a mouse model of candidiasis”. Lines274-275and 83-84.
12.Line 318: Change “leads to neutralize” to “neutralizes”
Ans:Now it is changed to “neutralizes” in line 32.
13.Line 334: delete “as”
Ans:“as” is deleted. Now that line is 341
414.Section 6. I would also add a fourth point that GlcNAc can induce signaling independently of its metabolism. This permits it to be studied as a signaling molecule independent of cellular metabolism.
Ans: We understand concern raised by you, and indeed it is a valid point.It is a very important finding that helped researchers to further study molecular details of GlcNAc signaling pathway. We have already mentioned this point (GlcNAc can induce signaling independently of its metabolism) at proper context within the text, line 90-97.,107-108. As per our understanding, either independency or dependency of GlcNAc signaling on its metabolism may not account as considerable aspect that would make C. albicans model system to study GlcNAc signaling.
15.Line 345: the commas are not needed.
Ans:The commas are removed. Now it is in line 352.
Reviewer 3 Report
This review describes the metabolism of GlcNAc in Candida albicans and the genes involved in the signal transduction system in relation to the aggravating factors of C. albicans infection.
A schematic diagram of metabolic system-related to catabolism after taking up GlcNAc into cells, which was pointed out during the previous review, was also written, and the latest papers were added to make it more complete. I think it has reached a level can be published in JoF.
Author Response
Comments and Suggestions for Authors
Q. This review describes the metabolism of GlcNAc in Candida albicans and the genes involved in the signal transduction system in relation to the aggravating factors of C. albicans infection.
A schematic diagram of metabolic system-related to catabolism after taking up GlcNAc into cells, which was pointed out during the previous review, was also written, and the latest papers were added to make it more complete. I think it has reached a level can be published in JoF.
Ans: The authors thank the anonymous reviewer for putting efforts in reading and providing us the valuable comments that helped us to improve the manuscript.
Round 2
Reviewer 2 Report
The authors have addressed many of my specific criticisms. However, there are still a number of issues, some of which I have listed below. There are also a lot of issues with the written document that need to be addressed. The authors should get help from a professional editor to improve the text.
- Lines 47-48.Not clear why the authors link GlcNAc induced cell death to colonizing a nutrient poor niche. GICD occurs under very specialized conditions that are not likely to be encountered in vivo.
- Line 169, 171, 195, 196, 212…363.The authors are still using the phrase “without inoculation”. This makes no sense to the reader. If you don’t inoculate then that means you are referring to sterile media. It’s not clear why the authors are not changing this to make it understandable.
- Line 226 – 230.The authors are still writing a confusing summary of the role of the cAMP pathway in GlcNAc responses. For example, on line 228, they state that the Ras1-cAMP-PKA pathway is needed for hyphal induction. To support this statement they cite a 22 year old paper that used inhibitors of PKA, and two review articles. One problem is that ref #43 showed that a ras1D mutant still induces hyphae in response to GlcNAc. Another problem is that reference #43 also provided data showing that cells lacking cAMP can still be induced to form hyphae. Thus, the authors have not presented any data to support a role for Ras1 and cAMP. This is true in other places in the manuscript too.
- Relating to the role of cAMP, it is interesting that Maidan et al. MBoC, 2005 reported that they did not detect an increase in cAMP levels after adding GlcNAc (although this was data not shown).
- Line 241.“It [rep1] is also shown to act as a transcription factor for galactose signaling apart from GlcNAc signaling [37].” The authors should restate this to say that Rep1 is also needed for growth on galactose. I don’t think there is evidence yet that it is acting as a transcription factor. Although this seems likely, it could be affecting galactose indirectly.
- Line 242: “Initially it was recognized as a negative regulator of MDR1 expression, the drug efflux pump [54].”This citation is for a paper about Ndt80, not Rep1. I think the authors intended to cite: Chen CG, et al. (2009) Rep1p negatively regulating MDR1 efflux pump involved in drug resistance in Candida albicans. Fungal Genet Biol 46(9):714-20
- Lines 243-247.Newer data indicates that Ron1 has a minor role in GlcNAc signaling. See ref.#56 by Min et al.
- Lines 247-249.This sentence about Ngs1 should be included earlier in the paragraph where Ngs1 is described.
- Lines 262 – 270.It would be interesting here to review the data that support the conclusion that GlcNAc stimulates White/Opaque switching through the cAMP pathway. For example, there seems that there is good data to support a role for Ras1 in inducing switching. This contrasts with hyphal induction, suggesting that the pathways are different.
- The binding of GlcNAc to Ngs1 has not been shown directly.The authors state that GlcNAc binds to Ngs1 in several places in the manuscript. It would be more helpful to the readers to state the evidence that supports GlcNAc binding to Ngs1 and review, and whether this conclusion is supported.
- Lines 273.The authors have still not explained in the text what they mean by GlcNAc priming.
- Line 338.Delete “or repressors (Hxk1)”. This does not fit in the context of the sentence.
Author Response
The authors have addressed many of my specific criticisms.However, there are still a number of issues, some of which I have listed below.There are also a lot of issues with the written document that need to be addressed.The authors should get help from a professional editor to improve the text.
Ans:In the revised version of the manuscript, we have taken care of editorial corrections apart from answering subject related issues raised by you. We have highlighted all the incorporated corrections with yellow.
1.Lines 47-48.Not clear why the authors link GlcNAc induced cell death to colonizing a nutrient poor niche.GICD occurs under very specialized conditions that are not likely to be encountered in vivo.
Ans: Here, we agree with your opinion. This was overemphasized from our end. Since the phenomenon of GICD is shown to occur under specialized conditions and is not verified under in vivo host niche environmental conditions, we have edited the sentence by removing “that provide the ability to the pathogen to colonize nutrient poor niche” (Line 47-48of previous version).
Checking GICD under in vivo conditions would probably be a potential area of future research to understand the mechanistic insights into the molecular mechanism of GICD and its significance inside the host.
2.Line 169, 171, 195, 196, 212...363.The authors are still using the phrase “without inoculation”. This makes no sense to the reader.If you don’t inoculate then that means you are referring to sterile media.It’s not clear why the authors are not changing this to make it understandable.
Ans:Yes, you are correct, the sentences may confuse the readers. Now, in the revised version of the manuscript we rectified the phrase in all the relevant places (lines 171, 173, 197, 198-199,214, 225-226, 362) to convey that “without fresh-medium inoculation”.
3.Line 226 –230.The authors are still writing a confusing summary of the role of the cAMP pathway in GlcNAc responses.For example, on line 228, they state that the Ras1-cAMP-PKA pathway is needed for hyphal induction.To support this statement they cite a 22 year old paper that used inhibitors of PKA, and two review articles.One problem is that ref #43 showed that a ras1D mutant still induces hyphae in response to GlcNAc.Another problem is that reference #43 also provided data showing that cells lacking cAMP can still be 2induced to form hyphae.Thus, the authors have not presented any data to support a role for Ras1 and cAMP.This is true in other places in the manuscript too.
Ans:To bring clarity,we have removed the confusing sentences related to the role of cAMP pathway in GlcNAc responses; lines 226-230(previous version)have been removed as“But whereas under the conditions when cells from a saturated culture are inoculated into a fresh medium, GlcNAc triggered filamentation appears to be dependent on Ras1-cAMP-PKA pathway [42, 48, 51]. Still, there are some reports indicating the involvement of additional pathways that likely to regulate GlcNAc-induced filamentation [43,58]”.
Also in the conclusions section,lines 366-369 are removed; as “But under the conditions of inoculation into a fresh medium, Ras1-cAMP/PKA signaling pathway acts as a major regulator of GlcNAc-induced morphological transitions including filamentation [42] and white to opaque switching”.
But these issues have been discussed in the relevant context in comparison with white/opaque switching. The role of Ras1-cAMP-PKAindependent pathways in GlcNAc induced filamentation has been described in lines 264-268 and 365-368,that appears to be contrasting with white to opaque switching triggering pathway.
4.Relating to the role of cAMP, it is interesting that Maidan et al. MBoC, 2005 reported that they did not detect an increase in cAMP levels after adding GlcNAc (although this was data not shown).
Ans:Yes, this is a supporting reference that indicates that GlcNAc induced filamentation does not require increased cAMP levels.
5.Line 241.“It [rep1] is also shown to act as a transcription factor for galactose signaling apart from GlcNAc signaling [37].”The authors should restate this to say that Rep1 is also needed for growth on galactose.I don’t think there is evidence yet that it is acting as a transcription factor.Although this seems likely, it could be affecting galactose indirectly.
Ans:As there is no direct evidence yet to show that Rep1 acts as a transcription factor for the growth on galactose, we have modified the sentence according to your suggestion. “It is also needed for growth on galactose apart from GlcNAc [37]”(line 239-240).
6.Line 242: “Initially it was recognized as a negative regulator of MDR1 expression, the drug efflux pump [54].”This citation is for a paper about Ndt80, not Rep1.I think the authors intended to cite:Chen CG, et al. (2009) Rep1p negatively regulating MDR1 efflux pump involved in drug resistance in Candida albicans. Fungal Genet Biol 46(9):714-20.
Ans:The authors thank anonymous reviewer for his/her suggestion. Now, we have replaced previous reference with “Chen CG, et al. (2009). Rep1p negatively regulating 3MDR1 efflux pump involved in drug resistance in Candida albicans. Fungal Genet Biol 46(9):714-20.”(lines 542-544)and highlighted with yellow.
7.Lines 243-247.Newer data indicates that Ron1 has a minor role in GlcNAc signaling.See ref.#56 by Min et al.
Ans:As per your suggestion and reference 56, we had tone down our sentence indicating minor role ofRon1 in GlcNAc signaling (Line241to 242). We have edited the sentenceas,“In C. albicans, in addition to Rep1, there is another Ndt80 domain transcriptional factor Ron1, which is also appeared to be needed for growth on GlcNAc [53]”.
And we have removed the following part of the sentence, “and has been shown to induce GlcNAc catabolic genes and Hyphal Specific Genes (HSG) as similar to GlcNAc sensor Ngs1. It can induce the formation of hyphae irrespective of the presence of GlcNAc[53]”.
8.Lines 247-249.This sentence about Ngs1 should be included earlier in the paragraph where Ngs1 is described.
Ans:As per your suggestion we have included role of Ngs1 on maltose dependent medium, in the relevant context in the “concluding remarks” section in Lines379 to 380.
9.Lines 262 –270.It would be interesting here to review the data that support the conclusion that GlcNAc stimulates White/Opaque switching through the cAMP pathway.For example, there seems that there is good data to support a role for Ras1 in inducing switching.This contrasts with hyphal induction, suggesting that the pathways are different.
Ans:The comparative description of signaling pathways triggered by GlcNActoinduce two important morphological transitions(white/opaque switching and yeast/hyphal transition)is an interesting aspect. While GlcNAc induced white/opaque switching is dependent ofRas1-cAMP/PKA pathway,yeast/hyphal transition is independent of Ras1-cAMP/PKA pathway. This contrasting nature of GlcNAc to utilize different pathways has been explained in Lines264-268, and also in lines 365-368.
10.The binding of GlcNAc to Ngs1 has not been shown directly.The authors state that GlcNAc binds to Ngs1 in several places in the manuscript.It would be more helpful to the readers to state the evidence that supports GlcNAc binding to Ngs1 and review, and whether this conclusion is supported.
Ans:The conclusion that GlcNAc binds to Ngs1 is supported by reference 37(Su et al., 2016). 4Su et al (2016, Nature communications) have shown that GlcNAc has the ability to bind to GlcNAc sensor Ngs1 andNgs1 contains binding sites for GlcNAc. In their study,they have performed GlcNAc binding assay and identified residues that can bind to GlcNAc.So,in thecurrent revised versionof the manuscript,we have citedthe reference in the relevantcontext as per your suggestioninLines127,155, 212, 254and331,and highlighted with yellow.
11.Lines 273.The authors have still not explained in the text what they mean by GlcNAc priming.
Ans:‘GlcNAc priming’ has been describedinLine 271-272and highlighted with yellow.
12.Line 338.Delete “or repressors (Hxk1)”.This does not fit in the context of the sentence.
Ans:Now in the revised version, “or repressors (Hxk1)” has been removed
This manuscript is a resubmission of an earlier submission. The following is a list of the peer review reports and author responses from that submission.
Round 1
Reviewer 1 Report
Manuscript “N-acetylglucosamine Signaling: Transcriptional dynamics of a novel Sugar Sensing Cascade in a model pathogenic yeast, Candida albicans” relates to the interesting issue of pathogens' detecting of GlcNAc and the influence of this sugar on basic metabolism and virulence mechanisms.
As this is a quite up-to-date topic, some reviews have been published recently, but this manuscript presents an interesting area related to transcription regulation. The work is quite well written and interesting, the figures complete the text, nonetheless I have some comments. Can the authors complete the information on white-opaque switching in addition to the information on morphological transition?
Moreover, the manuscript requires many stylistic and editorial corrections, as well as some language corrections. Authors should review all text carefully to correct these errors. Some examples are:
- all abbreviations of genes are to be italicized, such errors are numerous in the text and in the figures e.g. Figures 1 and 2
- Line 61 – the lack of N-
- Line 81 – vs. instead of Vs.
- Line 91 – the lack of figure number (Fig…..)
- Line 128 – into instead of in to
- degrees of temperature should be marked with the appropriate symbol
- some words do not have to be capitalized, e.g. serum, histone acetylation, white-opaque
- no additional captions of drawings are needed within them
- in Figure 3 the indication of A and B panels is missing
- in Figure 4 panels A and B should be marked instead of the left and right panels, already in the figure caption it is described in this way, changes should be made to the text
Reviewer 2 Report
This review addresses the interesting topic of how the sugar GlcNAc regulates transcription in the human fungal pathogen C. albicans. GlcNAc-regulated transcription is thought to contribute to phenotypic switches in C. albicans, including bud to hyphal growth and White/Opaque switching. C. albicans has become a leading model system for studying GlcNAc signaling since the model yeasts S. cerevisiae and S. pombe have lost the genes required to respond to metabolize or respond to this sugar. A strength of this review is that it focuses on transcription, which has not been the main focus of other recent reviews. However, there are major weaknesses. As described below, a whole section is devoted to a very speculative role for Hxk1 (GlcNAc kinase) in transcriptional regulation (which did not describe the alternative model that the hxk1D cells are simply responding to elevated levels of GlcNAc that accumulate when it cannot be metabolized due to the lack of the GlcNAc kinase). Another major oversight is that the review failed to include the synergistic role between GlcNAc and the Rim101 transcription factor. A third weakness is that there are many mistakes in the review, including citations to the incorrect references.
Major Comments:
- Section 4 on the role of Hxk1 (the GlcNAc kinase) in regulating gene expression should be extensively modified. The authors should cite other studies that concluded that an hxk1D mutant displays altered transcriptional responses because cells accumulate GlcNAc, not because Hxk1 plays a role in transcription. GlcNAc accumulates in hxk1Dmutant cells since it cannot be phosphorylated to GlcNAc -6-PO4 and metabolized. Thus, as hxk1D cultures grow, GlcNAc accumulates and induces signaling. This was confirmed by showing that hxk1D cells grown at low cell density do not display abnormalities attributed to the putative transcriptional role of Hxk1. See ref. 15. The authors can propose a possible role for Hxk1 in regulating transcription, but they should state that there are data that argue against it.
- The authors should mention the role of Rim101 in the regulation of GlcNAc transcriptional responses during hyphal formation.Cells growing on GlcNAc medium alkalize the extracellular pH, presumably because they excrete excess nitrogen as ammonia. This change in ambient pH acts through the Rim101 transcription factor to stimulate hyphal-induced genes (but not GlcNAc catabolic genes). See Naseem et al, 2015. PMID: 25609092
- There are very few references cited in sections 5, 6 and 7.More citations should be added for readers who want to learn more.
Minor comments
Line 43. How does GlcNAc Induced Cell Death (GICD) lead to the ability to colonize nutrient poor niche? This is unclear and should be corrected.
61 Missing the N (acetylglucosamine)
Line 91. precursor of UDP-GlcNAc that is constantly required for cell anabolic processes (Fig…),
Lines 95-98. As above, should be clearer that the altered phenotype of hxk1D cells is likely due to buildup of GlcNAc that can’t be metabolized.
Line 106 Unclear sentence. The mechanistic details were discussed in subsequent sections [14, 37].
Fig. 1 and Line 117: Rep1/Ron1 ???
Lines 121 and 141. Reference 38 is not correct in these places. Ref. 38 shows that GlcNAc can induce signaling independent of the known cAMP pathway, and is not related to transcriptional regulation.
Line 144: cAMP-PKA pathway does not have a role in induction of GlcNAc ???
Line 145-146: [18] is a wrong citation.
Line 146. What does “without inoculation” mean: (Su et al have demonstrated a mechanism for the stimulation of yeast to hyphal transition without inoculation in log phase cells at 37⁰C [18].
Same for lines 313-314???
A more recent study indicates that Ron1 does not play a major role in regulating GlcNAc in C. albicans. (see ref 39)
Line 217: It can induce the formation of hyphae irrespective of the presence of GlcNAc???
[38] is a wrong citation.
Lines 343-334. It is not clear that GlcNAc signaling is important for virulence. An ngt1D mutant does not display a virulence defect. What is known is that the abnormal cell signaling that occurs when cells can’t metabolize GlcNAc can affect virulence. Thus, “abnormal” GlcNAc signaling can impact virulence, but GlcNAc signaling is not required for virulence.
Line 344: This question has been answered in part by studies published by Min et al (see reference 39)
Ref. 16 and 20 have quotation marks in the title.
Ref. 20: no page number. wrong style.
Reviewer 3 Report
This review provides an easy to understand the cascade of genes involved in signal transduction associated with GlcNAc one of exacerbators of Candida albicans and those associated with metabolism as amino sugars merely as nutrient sources. In addition, the comparison between the GlcNAc signal transduction system in E. coli and that in Candida makes it easy to see the differences.
Regarding the cited documents, the main documents are listed without any bias, and the whole is organized in a well-balanced format.
However, it is recommended that the points be described more carefully, as there are several points that are not sufficiently detailed.
For example, in the description of the assimilation of UDP-GlcNAc on line 91, there is no corresponding figure, and it is (Fig...). It is necessary to add a simple figure and enter the Fig number.
In line 227, ADH1 is written twice, so it needs to be corrected.
I also recommend citing a review of GlcNAc sensing in the August issue of JoF.